# Glutamate-Mediated Excitotoxicity in the Pathogenesis and Treatment of Neurodevelopmental and Adult Mental Disorders

**DOI:** 10.3390/ijms25126521

**Published:** 2024-06-13

**Authors:** Noemi Nicosia, Mattia Giovenzana, Paulina Misztak, Jessica Mingardi, Laura Musazzi

**Affiliations:** 1School of Medicine and Surgery, University of Milano-Bicocca, 20900 Monza, Italy; n.nicosia@campus.unimib.it (N.N.); m.giovenzana6@campus.unimib.it (M.G.); paulina.misztak@unimib.it (P.M.); 2PhD Program in Neuroscience, School of Medicine and Surgery, University of Milano-Bicocca, 20900 Monza, Italy; 3Fondazione IRCCS San Gerardo dei Tintori, 20900 Monza, Italy

**Keywords:** glutamate, excitotoxicity, autism spectrum disorder, substance use disorders, schizophrenia, depression, therapeutics

## Abstract

Glutamate is the main excitatory neurotransmitter in the brain wherein it controls cognitive functional domains and mood. Indeed, brain areas involved in memory formation and consolidation as well as in fear and emotional processing, such as the hippocampus, prefrontal cortex, and amygdala, are predominantly glutamatergic. To ensure the physiological activity of the brain, glutamatergic transmission is finely tuned at synaptic sites. Disruption of the mechanisms responsible for glutamate homeostasis may result in the accumulation of excessive glutamate levels, which in turn leads to increased calcium levels, mitochondrial abnormalities, oxidative stress, and eventually cell atrophy and death. This condition is known as glutamate-induced excitotoxicity and is considered as a pathogenic mechanism in several diseases of the central nervous system, including neurodevelopmental, substance abuse, and psychiatric disorders. On the other hand, these disorders share neuroplasticity impairments in glutamatergic brain areas, which are accompanied by structural remodeling of glutamatergic neurons. In the current narrative review, we will summarize the role of glutamate-induced excitotoxicity in both the pathophysiology and therapeutic interventions of neurodevelopmental and adult mental diseases with a focus on autism spectrum disorders, substance abuse, and psychiatric disorders. Indeed, glutamatergic drugs are under preclinical and clinical development for the treatment of different mental diseases that share glutamatergic neuroplasticity dysfunctions. Although clinical evidence is still limited and more studies are required, the regulation of glutamate homeostasis is attracting attention as a potential crucial target for the control of brain diseases.

## 1. Introduction

Glutamate (or glutamic acid) is a positively charged non-essential amino acid which, besides its metabolic roles [1], also represents the most important excitatory neurotransmitter in the central nervous system (CNS), where it drives approximately 70% of synapses [2]. The neurotransmitter glutamate regulates various brain processes ranging from cognitive functions, such as learning and memory, to mood, control of the sleep–wake cycle, pain perception, and motor function [3,4,5]. Glutamatergic transmission is mediated by specific ionotropic and metabotropic receptors located both at postsynaptic and presynaptic terminals. Ionotropic receptors, including AMPA (α-amino-3-hydroxy-5-methyl-a-isoxazolepropionate), NMDA (N-methyl-d-aspartate), and kainate receptors, are nonselective cation channels, allowing the passage of sodium and potassium ions and, in some cases, small amounts of calcium ions. They are primarily expressed at postsynaptic sites, and their activation produces excitatory currents. The interaction of glutamate with AMPA and kainate receptors induces the generation of a very rapid but weak excitatory postsynaptic current [6], which, however, is enough to allow the removal of the Mg^2+^-dependent block of the NMDA receptor. In turn, the binding of glutamate and of the co-agonist glycine (or d-serine) promote the opening of NMDA receptors, which are permeable also to calcium ions, resulting in long-lasting postsynaptic currents [7]. In addition to ionotropic receptors, glutamate also acts through three groups of G protein-coupled metabotropic glutamate receptors (mGluRs), which modulate synaptic function more slowly than ionotropic receptors [8]. Group I includes mGlu1 and 5, Group II includes mGlu2 and 3, and Group III includes mGlu4, 6, 7, and 8. Group I is coupled to Gq/G11 and activates phospholipase Cβ, while group II and III are coupled predominantly to Gi/o proteins, thus leading to the inhibition of adenylyl cyclase and direct regulation of ion channels. Group I mGluRs are often localized postsynaptically, and their activation leads to cell depolarization and an increase in neuronal excitability. In contrast, group II and group III mGluRs are typically localized on presynaptic terminals, where they inhibit neurotransmitter release [9]. For a detailed description of ionotropic and metabotropic glutamatergic receptor signaling, activation mechanisms, and crosstalk, please refer to [10].

Glutamate dynamics play a critical role in maintaining glutamate homeostasis and are tightly regulated by neuron–astrocyte crosstalk [2]. Indeed, glutamatergic synapses are defined as “tripartite” as they are composed of a glutamate presynaptic terminal, a postsynaptic spine, and an astrocyte [11]. Since there are no enzymes able to metabolize glutamate in the extracellular space, excess glutamate is cleared from the extracellular space by neuronal and astrocytic excitatory amino acid transporters (EAAT1-5) [2]. In astrocytes, glutamate is converted by glutamine synthetase to glutamine, which is then released and uptaken by neurons, where it can be used to produce new glutamate. Glutamate is also released into extra-synaptic space by astrocytes via the cystine–glutamate antiporter, which thus contributes to the regulation of extracellular glutamate levels [12].

A dysfunctional alteration of glutamate homeostasis may lead to glutamate excitotoxicity, a condition in which the synaptic concentration of glutamate can reach up to 100 µM, leading to neurotoxicity and neuronal atrophy/death [13]. This condition may arise in the presence of (1) an impaired glutamate/glutamine recycle system, (2) dysfunctional reuptake and (3) altered expression and activity of glutamate receptors. An excessive synaptic glutamate concentration causes an overflow of intracellular calcium ions, which in turn leads to the activation of a transduction cascade [14]. In particular, excessive intracellular calcium levels lead to considerable excitotoxic damage by an alteration of the permeability of the mitochondrial membrane, resulting in the disruption of cell energy production, and by activation of calcium-dependent enzymes such as calpains, death-associated protein kinase 1 (DAPK1), and neuronal nitric oxide synthase (nNOS) with a consequent increase in production of nitric oxide [14,15]. This, in turn, leads to an increase in the production of reactive oxygen (ROS) species that, together with the hyperactivation of proteases and lipases, contribute to oxidative stress and cell death by apoptosis [14].

Glutamate excitotoxicity has received attention as a putative mechanism in the etiopathogenesis of several CNS disorders, including neurodevelopmental, neurological, neurodegenerative, and mental diseases [16,17,18,19,20].

The aim of this narrative review is to summarize evidence implying mechanisms of glutamate excitotoxicity in the pathophysiological mechanisms and therapy of neurodevelopmental disorders, with a focus on autism spectrum disorders (ASD), and adult mental disorders, with particular attention to substance abuse and psychiatric disorders. Among others, we selected these diseases because they have in common recent therapeutic approaches that include glutamatergic drugs, thus suggesting that direct modulation of glutamatergic transmission may exert therapeutic effects in these conditions.

Data for this review were collected using the PubMed database.

## 2. Involvement of Glutamate Excitotoxicity in Autism Spectrum Disorders

ASDs include a complex family of neurodevelopmental disorders characterized by repetitive behaviors, abnormalities in communication, and impaired social interaction skills, frequently together with other concurrent medical or psychiatric conditions [21,22].

DSM V defines ASD as a condition with “persistent deficits in social communication and social interaction across multiple contexts” and “restricted, repetitive patterns of behavior, interests, or activities” from early developmental phases and significantly impacting daily life [23].

Compelling evidence has highlighted a role for glutamate-dependent excitotoxicity in the pathophysiology of ASDs. Indeed, clinical studies consistently reported increased glutamate levels and receptor subunits in both the serum and brain areas of ASD patients, and a prolonged imbalance between excitatory and inhibitory transmissions was involved in etiopathogenetic processes [24,25,26]. Indeed, an overexcitation (or weak inhibition) of cortical function has been associated with a broad range of abnormalities in perception, memory, cognition, and motor control [27]. Intriguingly, genetic evidence has also clearly implicated glutamate receptors and transporter systems in ASD, since several polymorphisms of glutamate receptor subunits and glutamate transporter genes have been associated with the disease [28]. Moreover, rare mutations in structural proteins of the postsynaptic density regulating glutamatergic transmission, such as Shank proteins, have also been associated with ASD [29].

Accordingly, several animal models of ASD, including both pharmacological models and genetic models generated by mutations in genes found in patients with ASD, show glutamatergic alterations. Excitotoxicity, altered glutamate homeostasis, and receptor subunit expression and regulation were reported in the brain of a valproic acid model, one of the most widely used animal models of ASD [30,31,32,33]. Similarly, Shank mutant mice clearly exhibit glutamatergic impairments [34,35,36,37]. In both models, glutamatergic interventions can rescue ASD-like behavioral dysfunctions [35,37,38].

### Glutamatergic Drugs for the Treatment of Autism Spectrum Disorders

Different glutamatergic approaches were considered for the management of ASDs.

Riluzole, an inhibitor of voltage-dependent sodium channels and NMDA/kainate receptor antagonist, and memantine, a non-selective NMDA receptor antagonist, were shown to improve ASD symptoms in both children and adult patients [39,40]. The therapeutic properties of low-dose intranasal ketamine, another NMDA receptor antagonist approved as an anesthetic and recently introduced at subanesthetic dose as a rapid-acting antidepressant (see Section 5.1), were also tested in adolescent and young adults with ASD, showing some therapeutic potential with limited and transient adverse effects [41,42]. Nevertheless, the clinical evidence is still very limited, and more studies are required to understand whether glutamatergic agents can be safe and effective therapeutic strategies for ASD.

mGluR5 antagonists have been the only class of glutamatergic modulators that has made significant advances in drug development for the treatment of ASD. However, despite the enthusiasm raised due to their success in preclinical phases, unfortunately mGluR5 antagonists have failed in phase III clinical trials on ASD patients for lack of efficacy [43,44].

Finally, another molecule with multiple actions, including antioxidant effects, reduction in cytokine activity, modulation of dopamine release, reversal of mitochondrial dysfunction, reduction in apoptosis, anti-inflammatory activity, increased neurogenesis, and regulation of glutamate homeostasis, has shown promising effects in ASD [45]. It is N-acetyl-l-cysteine (NAC), the acetylated precursor of l-cysteine, which has been shown to reduce hyperactivity and irritability and to enhance social awareness in ASD [46]. However, future trials with larger sample sizes, confounding effects controlled, and long-term follow-up are warranted.

## 3. Involvement of Glutamate Excitotoxicity in Substance Use Disorders

As defined by the DSM-5, substance-related disorders (SRDs) and addictive disorders encompass a spectrum of treatable mental health conditions attributed to the problematic consumption of various substances, such as alcohol, opioids, stimulants, cannabis, hallucinogens, inhalants, and others [23]. Glutamate is an important mediator in both the neurotoxicity induced by abused drugs and the development and maintenance of addiction. Indeed, the rewarding process implies the intricate and bidirectional interplay between glutamate and dopamine transmissions within the mesolimbic pathway [47]. In response to rewarding stimuli, glutamate modulates dopamine release by activating NMDA and AMPA receptors on dopaminergic neurons in the ventral tegmental area (VTA). As a result, dopaminergic inputs from VTA to glutamatergic neurons in the prefrontal cortex (PFC), amygdala, and hippocampus induce alterations of glutamate release and receptor expression [48]. Consequently, drug abuse induces dynamic alterations in plasticity mechanisms such as long-term potentiation (LTP) and long-term depression (LTD) in corticolimbic brain areas, in turn impacting processes related to mood, learning, and memory [49].

In this section, we will focus on selected classes of abused drugs which cause alterations of the glutamatergic system at different stages of exposure and withdrawal, focusing on works where glutamate-mediated excitotoxicity has been directly implicated.

### 3.1. Cocaine

Cocaine remains a prominent choice among drugs of abuse worldwide, renowned for its potent psychostimulant properties and its ability to induce euphoria. Similarly to other psychostimulants, cocaine induces its effects primarily through activation of the mesocorticolimbic dopaminergic system [50]. Nevertheless, a growing number of preclinical studies shed light on the critical role of glutamate homeostasis in neurotoxicity at different stages of cocaine exposure [51]. In fact, acute cocaine exposure of rats affected glutamate levels within VTA [52,53], whereas a challenge dose augmented its release in medial PFC (mPFC) after repeated treatment [54,55]. In line with these findings, further research employing microdialysis showed that glutamate levels increased within the rat striatum [56] and nucleus accumbens [57] after a challenge dose. Conversely, rats exposed to 20 days of cocaine self-administration exhibited a notable reduction in extracellular glutamate levels in the nucleus accumbens, then reversed upon the administration of a priming dose of the drug [58]. In addition, chronic cocaine exposure has been associated with changes in the expression levels of both ionotropic and metabotropic glutamate receptors within the mPFC in rats [59] and the caudate nucleus of non-human primates [60].

### 3.2. Amphetamines

Alongside cocaine, amphetamine (AMPH) and its derivatives including 3,4-methylenedioxymethamphetamine (MDMA) are widely recognized as some of the most misused psychostimulants, especially among adolescents. These substances trigger an excessive release of monoamines at the synaptic level by binding membrane-located transporters of dopamine [61], norepinephrine, and serotonin [62] but also lead to a sustained rise in glutamate, leading to neurotoxicity [63]. Studies on rats showed that repeated exposure to methamphetamine augmented glutamate levels within various brain regions including the striatum [64], hippocampus [65], VTA [66], nucleus accumbens, and PFC [67]. Accordingly, amphetamines also induced significant alterations in the expression of AMPA receptor subunits GluA1 and GluA2 in the nucleus accumbens of rats undergoing a repeated treatment [68]. Moreover, MK-801 prevented chronic methamphetamine-induced neurotoxicity by blocking NMDA receptors [69], while repeated exposure worsens neuronal damage in the striatum through mechanisms involving AMPA receptor-mediated excitotoxicity and calpain-specific spectrin proteolysis, a process implicated in cellular apoptosis [70]. In addition, further evidence from in vivo studies supports the involvement of the nitric oxide pathway [71] and the tumor necrosis factor (TNF)-α [72] in methamphetamine-induced excitotoxicity.

Similarly to other amphetamines, MDMA, commonly known as ecstasy, has been shown to promote a sustained increase in extracellular glutamate release within the hippocampus in rats, which was then attenuated by treatment with ketanserin/fluoxetine [73] and ketoprofen [74]. Furthermore, MDMA reduces the parvalbumin-positive gamma-aminobutyric acid (GABA)ergic neurons in the hippocampus, suggesting that MDMA-induced damage involves an excitatory/inhibitory imbalance [74]. Intriguingly, some findings point to the critical involvement of glial cell activation in the neuronal damage and neurotoxic effects induced by MDMA exposure. Indeed, MDMA exposure led to microglia activation in mouse striatum, resulting in neuronal death along with an increased expression of glial fibrillary acidic protein (GFAP) [75].

### 3.3. Ketamine

Since its introduction in the 1960s as a dissociative anesthetic, ketamine has drawn attention for its recreational use due to its dissociative effects. Ketamine acts as a non-competitive antagonist of NMDA receptors primarily targeting inhibitory neurons in corticolimbic brain areas, thereby amplifying glutamatergic transmission [76]. Accordingly, daily exposure to ketamine was reported to cause glutamate-induced neurotoxicity, oxidative stress, and apoptosis in both the cerebral cortex and hippocampus [77,78]. Moreover, prolonged ketamine exposure was reported to enhance NMDA receptor expression and ROS production in rat primary forebrain cultures, ultimately culminating in neuronal death [79,80].

It is also worth mentioning that early prenatal or postnatal exposure to ketamine as an anesthetic can induce neurodevelopmental alterations leading to long-term cognitive impairment and learning deficits [81]. This has been associated with an overexpression of NMDA receptors, mitochondrial dysfunction, oxidative stress, and defects in neurogenetic pathways [82].

### 3.4. Ethanol

Ethanol, the primary psychoactive compound found in alcoholic beverages, is responsible for intoxicating effects, and its overconsumption represents a serious societal challenge, significantly contributing to worldwide morbidity [83]. Ethanol passes through the blood–brain barrier (BBB) and potentiates GABAergic transmission, thus affecting excitatory/inhibitory balance [84]. Importantly, alcohol also directly modulates excitatory glutamatergic neurotransmission. Indeed, while acute ethanol intake induces the downregulation of postsynaptic NMDA receptors, chronic exposure increases the expression of these receptors resulting in excitotoxic cascade events, including neuronal death [83]. Post-mortem analyses of brains from individuals with a history of alcohol abuse showed that ethanol further potentiates glutamatergic transmission in several brain regions by decreasing the expression of GLT-1 transporters [85,86] and damaging astrocytes [87], which are responsible for at least 90% of glutamate reuptake [83,88,89,90].

Evidence from preclinical studies have provided insights into the cellular neurobiological mechanisms implied in ethanol induced excitotoxicity. For instance, in vitro chronic ethanol exposure was reported to determine abnormal synaptic transmission and excitotoxic effects in hippocampal slices involving the dysregulation of ionotropic and metabotropic glutamate receptor subunit expression [91,92]. On the other hand, chronic alcohol exposure in rats has been shown to trigger calpain activity both in the cerebral cortex and cerebellum, confirming its role in cellular damage induced by alcohol consumption [93]. Furthermore, chronic alcohol exposure was also reported to increase the levels of the stress hormone glucocorticoid, which, in turn, overactivate NMDA receptors, particularly the NR2A and NR2B subunits, exacerbating neuronal cell death induced by alcohol excitotoxicity [83].

Importantly, ethanol exposure during early neurodevelopmental phases dramatically impacts synaptogenesis and triggers massive neuronal apoptosis, that in turn may contribute to mental disability syndromes including fetal alcohol spectrum disorders (reviewed in [94]). In addition, combined abuse of ethanol and ketamine or methamphetamine has been shown to synergically exacerbate glutamate-induced excitotoxicity, leading to neurotoxic consequences [95,96].

### 3.5. Therapeutic Approaches for Substance Use Disorders Based on the Rescue of Glutamate-Induced Neurotoxicity

Despite the valuable insight provided by existing research on excitotoxicity induced by substance-related disorders, further investigations are needed to fully clarify these mechanisms in order to identify potential targets for therapeutic interventions. Nevertheless, given the crucial role of glutamate in the pathophysiology of addiction, therapeutic strategies targeting glutamate-induced neurotoxicity hold promise for the treatment of substance-related disorders. In this regard, cation channel blockers (Lamotrigine) and NMDA receptor antagonists (Acamprosate, Amantadine, Memantine, and MK-801) have gained much consideration in the development of therapeutic interventions, since they have been reported to protect neurons from excitotoxicity [88,97,98,99]. Among these, memantine and MK-801 have shown efficacy in reducing cocaine and alcohol craving and relapse in preclinical models by blocking excessive calcium influx through NMDA receptors [100,101,102,103,104]. However, the clinical evidence is conflicting and inconclusive [105]. On the other hand, AMPA receptor antagonists such as topiramate have been more successful in showing neuroprotective properties against glutamate neurotoxicity [61,106] and have demonstrated efficacy in reducing alcohol intake and preventing relapse in human subjects [107] and increasing abstinence in cocaine consumers [108].

Moreover, preclinical studies have demonstrated that cannabidiol, an active compound found in cannabis, exerts anti-addictive properties (reviewed in [109,110]). Interestingly, cannabidiol attenuates excitotoxicity induced by various drugs of abuse by inhibiting the release of glutamate and reducing the activity of NMDA receptors. Therefore, through the modulation of glutamatergic neurotransmission, cannabinoids may help in preventing excessive calcium influx into neurons with protective effects against neuronal damage [111]. Cannabidiol is currently under investigation in clinical trials for the treatment of alcohol abuse [110].

Finally, serotonergic psychedelics, thanks to their neuroplastic action and capability to increase brain network connectivity, are under study as new treatment options for substance use disorders [112,113]. Although classic hallucinogens exert their pharmacological effects primarily through the serotonergic system, acting as agonists of the 5-HT_2A_ receptor, their action goes beyond this target and also involves the modulation of glutamatergic transmission (see Section 5.1).

## 4. Glutamatergic Alterations in Schizophrenia

Schizophrenia is a chronic psychiatric disorder with a prevalence estimated around 1% globally and is characterized by a broad symptomatology extending from positive (i.e., hallucinations and delusion) to negative symptoms (i.e., anhedonia and asociality) and cognitive deficits, thus severely impacting on the life of patients and their families [20]. Although the first theory of schizophrenia, derived from evidence of antipsychotic effects of antidopaminergic drugs, is centered on dopaminergic dysfunction, a role for the glutamate system has been speculated since the observation that NMDA receptor antagonists like ketamine or phencyclidine induced a schizophrenic-like state in healthy subjects and an aggravation of symptoms in schizophrenic patients [20,114,115]. Of note, the psychotic effects of NMDA receptor antagonists persist in the absence of dopamine activity or treatment with dopamine antagonists, further highlighting the importance of glutamatergic alterations in the induction of psychotic symptoms [116,117].

Although the NMDA receptor antagonists may be believed to cause a hypofunction of glutamatergic transmission, the fact that ketamine and phencyclidine have high affinity for NMDA receptors localized on GABAergic interneurons gives reason to the induction of increased glutamatergic transmission through disinhibition processes [118,119,120]. This leads to glutamate hyperexcitability, downstream stimulation of the mesolimbic pathway, and schizophrenic symptoms. At the same time, several lines of evidence have shown the importance of an excitatory/inhibitory imbalance in schizophrenia pathophysiology. Indeed, GABAergic disfunctions were reported in different brain areas of patients [117].

Preclinical models of schizophrenia have been based on the treatment of rodents with phencyclidine/phencyclidine-like drugs or on genetically modified animals with impaired glutamate receptor expression/function in specific brain areas (i.e., NMDA receptor subunit zeta1 knockdown, epsilon1 knockout, and zeta1 point mutant mice) [118,121]. Importantly, several studies using both mice and rats reported increased extracellular glutamate levels in cortical and subcortical regions following injection with phencyclidine or ketamine. Early studies reported neuronal vacuolization and necrosis induced by both phencyclidine and MK-801 [122]. Interestingly, low doses induced reversible vacuolar changes, while higher doses or a prolonged treatment with both phencyclidine or MK-801 produced irreversible and more widespread damages [123,124]. Furthermore, phencyclidine was shown to upregulate the hsp70 stress gene in the cortex, hippocampus, and basal nuclei of the amygdala of rats, contributing to increased intracellular calcium levels and subsequent apoptotic processes [125].

Interestingly, Schobel and colleagues demonstrated that excessive glutamate levels in specific subregions of the hippocampus in mice following repeated administration of ketamine were associated with neuronal hypermetabolism and could be responsible for consequent neuronal atrophy [126]. Importantly, in the same study, the authors showed that subjects at clinical high-risk for a psychotic disorder showed hypermetabolism in the hippocampus [126].

Moreover, imaging studies on schizophrenic patients consistently reported volumetric alterations of glutamatergic cortical and non-cortical brain regions together with white matter abnormalities [127] and changes in glutamate levels [118], thus confirming that glutamatergic dysfunction plays a role in the functional and morphological changes underlying schizophrenia. Importantly, clinical evidence strongly supports the hypothesis that the volumetric reductions and cortical thinning observed in schizophrenic patients may be related to neuroanatomical compromise through an excitotoxic process [128].

Further evidence linking psychosis with glutamatergic alterations comes from anti-NMDA receptor encephalitis, a now well-established autoimmune disorder presenting with schizophrenia-like symptoms caused by autoantibodies against the NMDA receptor [129]. This leads to isolated psychotic presentations which efficiently respond to immunotherapies.

### Therapeutic Approaches for Schizophrenia Based on Glutamatergic Approaches

Low levels of the NMDA receptor co-agonist d-serine were found in the cerebrospinal fluid and post-mortem brains of schizophrenic patients, suggesting a possible functional contribution to NMDA receptor hypofunction. Moreover, schizophrenic patients were shown to have higher levels of d-amino acid oxidase, the peroxisomal flavoenzyme responsible for the metabolism of d-serine [130,131]. Early clinical trials employing d-serine as a supplementary treatment to antipsychotics showed significant improvements in positive, negative, and cognitive symptoms in schizophrenic patients. Although larger-scale studies have obtained conflicting results, more phase II and III trials are ongoing [132]. Moreover, both preclinical and clinical studies have considered the co-administration of d-serine analogs together with d-amino acid oxidase inhibitors, such as Luvadaxistat and sodium benzoate, as therapeutic approaches to enhance NMDA receptor function in schizophrenia [133]. In a randomized, double-blind, placebo-controlled study, Lin and colleagues observed a beneficial therapeutic effect of the addition of sodium benzoate to clozapine in the treatment of schizophrenic patients which positively correlated with antioxidant effects [134]. Yet, these results have not been replicated in larger studies. Moreover, preclinical studies have shown that the potential antioxidant properties of d-amino acid oxidase inhibitors might be dependent on the dosage because high doses are associated with increased oxidative stress [131,135,136].

Inhibitors of the glycine transporter-1 (GlyT1), which is responsible for glycine reuptake from the synaptic space, have been developed as well and tested for the treatment of schizophrenia. Sarcosine, an endogenous amino acid analog to d-serine acting both as a competitive inhibitor of GlyT1 and as a co-agonist of the NMDA receptor, has shown promising therapeutic benefits when combined with antipsychotic medication [137,138,139]. Iclepertin (BI 425809) is a selective GlyT1 inhibitor currently located in clinical phase III that showed pro-cognitive effects in patients with schizophrenia [140] and memory-enhancing effects in rodent cognition tasks, together with a decrease in the deficit in EEG parameters induced by MK-801 [141].

Another strategy adopted to compensate for the NMDA receptor hypofunction observed in schizophrenia is the positive modulation of the metabotropic glutamate receptors mGluR5 [142,143]. Indeed, post-mortem studies have reported reduced expression of mGluR5 in brain areas of schizophrenic patients compared to healthy controls, suggesting an involvement in pathophysiological processes [142,143,144].

Preclinical evidence has shown that mGluR5 positive allosteric modulators ameliorate cognitive impairment and negative symptoms in NMDA antagonist-induced models [145]. It is important to underline that, since in addition to directly increasing cellular excitability, mGluR5 physically interacts with NMDA receptor subunits to improve their activity, the use of positive allosteric modulators instead of full mGluR5 agonists seems to avoid the risk of excitotoxicity due to direct and prolonged activation of glutamate receptors [146]. However, the therapeutic potential of mGluR5 positive allosteric modulators has not yet been tested clinically.

Finally, mGluR2/3 agonists are also under evaluation for their possible antipsychotic effect. Preclinical studies have already provided exciting results, and pomaglumetad has entered clinical phase development. Pomaglumetad has been considered both in monotherapy and in combination with other treatments, although inconsistent results were observed in both [132]. A recent exploratory analysis suggested that pomaglumetad might be more effective in the early stages of the disease [147].

## 5. Glutamatergic Alterations in Major Depressive Disorder and Post-Traumatic Stress Disorder

Stress-related mental disorders, such as major depressive disorder (MDD) and post-traumatic stress disorder (PTSD), are common mental health conditions that severely impact patients’ well-being and represent a global burden on society [148]. Patients suffering from MDD experience persistent low mood and anhedonia eventually associated with metabolic dysfunctions, sleep disturbances, cognitive impairments, and increased suicidal risk. On the other hand, PTSD is mostly characterized by dysfunctional processing of fear associated with recurrence of intrusive memories, hyperarousal and/or avoidance, and emotional distancing. Convergent post-mortem and brain imaging studies have indicated morphological and functional changes in the brain of both MDD and PTSD patients, particularly in areas where glutamatergic transmission is predominant [19,149,150,151]. Reduced volume and decreased activity of HPC and PFC were reported in MDD and were associated with disease severity and lack of therapeutic response [149]. Conversely, amygdala and nucleus accumbens volume and function have been found to be increased in MDD patients [152,153]. Similarly, human studies reported alterations of the excitatory/inhibitory balance in corticolimbic areas of PTSD patients [19], as well as HPC and PFC volume reductions and increased amygdala volume in some patients [151]. Overall, this evidence strongly supports the importance of glutamatergic transmission in the etiopathogenetic process of stress-related disorders.

Accordingly, preclinical models based on stress exposure strongly support that stress dramatically affects glutamatergic transmission in the same brain areas affected in patients [154]. As for HPC and PFC, acute stress has been consistently shown to rapidly enhance glutamatergic transmission by increasing glutamate release and inducing changes in glutamate receptor activation and trafficking (reviewed in [3,155]). Conversely, chronic exposure to stress has been mostly associated with deficits in glutamatergic transmission in PFC and HPC [154,156,157]. Indeed, chronic stress was shown to induce changes in the levels and function of both ionotropic and metabotropic glutamate receptors, leading to a reduction in synaptic strength and participation in dysregulated glutamate release [154,156]. Preclinical models based on both acute and chronic stress have also shown neuronal dendritic simplification and spine reductions in HPC and PFC, implying that glutamatergic dysfunctions may underlie the architectural abnormalities measured in patients [152,158]. Importantly, functional and morphological alterations induced by chronic stress were selectively observed in vulnerable animals and not resilient ones, confirming that the glutamatergic changes play a role in the maladaptive response to chronic stress [159,160,161]. Although the specific mechanisms causing functional impairment and dendritic simplification in corticolimbic areas under chronic stress exposure have not yet been identified, a main hypothesis implicates glutamatergic transmission [154,156]. Indeed, it has been speculated that the increase in glutamate release and transmission rapidly induced by stress may lead, in the long term, to adaptive excitotoxic processes, in turn causing a reduction in synaptic density and strength. Accordingly, rapid-acting antidepressants induce a transient surge in prefrontal glutamatergic neurotransmission, which is associated with rapid and sustained cortical connectivity (see Section 5.1).

Differently, studies focusing on the amygdala and nucleus accumbens mostly agree in reporting that both acute and chronic stress increase glutamatergic transmission, mainly in the basolateral component that receives connections from the PFC, leading to the consolidation of emotional memories [162,163]. Indeed, the amygdala is a central hub integrating and processing information related to fear and anxiety, with important implications in the regulation of memory, motivation, and autonomic responses [162].

Finally, it is worth noting that dysfunctional glutamatergic signaling was also registered in adult animals that were exposed to early life stress or corticosterone administration in adolescence, suggesting that stress-induced glutamatergic changes may be long-lasting and contribute to the pathogenesis of depressive disorders later in life [164,165].

### 5.1. Glutamatergic Rapid-Acting Antidepressant Drugs

Traditional antidepressant drugs, such as selective serotonin reuptake inhibitors (SSRIs), primarily acting by increasing the synaptic bioavailability of monoamines, were also shown to modulate glutamatergic transmission [166,167]. Indeed, monoaminergic antidepressants were demonstrated to regulate the expression and function of NMDA and AMPA receptors, to modulate glutamatergic synaptic transmission, and to rescue morphological changes in corticolimbic brain areas [168,169,170].

Accordingly, accumulated knowledge on the etiopathogenetic processes of MDD and PTSD converge in attributing a major role to neuroplasticity impairments and dysfunction of the glutamatergic system [171]. In this context, the evidence of rapid and long-lasting antidepressant properties of the non-competitive NMDA receptor antagonist ketamine administered at subanesthetic dose revolutionized research on antidepressants, which had not seen drugs with new mechanisms of action for decades [156]. Indeed, even though traditional antidepressants are effective in most patients, their efficacy is limited by a delay of therapeutic onset of several weeks and by a high percentage of non-response and disease recurrence [172]. Preclinical studies have revealed that the rapid antidepressant effect of ketamine is associated with the restoration of dysfunctional glutamatergic transmission and promotion of neuroplasticity in corticolimbic areas [150,173,174,175].

This amount of research led to the authorization of an intranasal formulation of esketamine, the (S)-enantiomer of ketamine, for the management of treatment-resistant (TRD) patients in combination with a classic antidepressant in 2019 [176], and ketamine is also receiving attention for the treatment of PTSD and other psychiatric conditions [177,178].

Despite the enthusiasm raised by the rapid psychotropic action of ketamine, the abuse potential and other possible adverse effects represent significant limitations on wider use of the drug in therapy. Nevertheless, the success of ketamine has supported the investigation of antidepressant properties of other glutamatergic drugs.

Several glutamatergic drugs including selective antagonists of the GluN2B subunit (CP-101606; MK-0657), NMDA receptor partial agonists (d-Cycloserine), and glycine site modulators (GLYX-13) have been investigated in recent years [172,179]. However, all these drugs have failed in clinical trials due to a lack of antidepressant efficacy in patients and have been thus abandoned. Other glutamatergic drugs still under preclinical/early clinical development include the following: R-ketamine, hydroxynorketamine, modulators of metabotropic receptors (especially mGluR2/3 antagonists), dextromethorphan, dextromethadone, and nitrous oxide [177,179,180,181].

At the same time, since ketamine is a dissociative psychedelic drug, it has also reignited interest of research in unveiling the therapeutic potential of other psychedelic drugs. Indeed, hallucinogens like psilocybin, LSD, and MDMA are being studied not only as innovative, fast-acting antidepressants but also as drugs for the treatment of alcohol and substance use disorders, as well as for the management of PTSD [182,183,184,185]. Importantly, despite primarily targeting serotonergic transmission, most psychedelics were reported to downstream modulate glutamatergic transmission as a mechanism to promote neuroplasticity [186,187]. Indeed, psychedelics were shown to increase glutamate release and transmission predominantly in cortical brain regions, in turn activating neurotrophic cascades, inducing long-term structural plasticity and modifications of cortical functional connectivity [188].

## 6. Discussion and Future Perspectives

For decades, neurodevelopmental and adult mental disorders have been studied in parallel, with limited consideration of possible common etiopathogenetic factors and therapeutic strategies. The same happened with neuropsychiatric and substance use disorders. However, convergent recent evidence is highlighting the importance of glutamatergic regulation in all these clinical conditions. Impaired glutamatergic transmission may lead to excitation/inhibition imbalance, which possibly may activate excitotoxic processes, in turn affecting neuroplasticity and eventually neuronal survival [13]. Indeed, even though excitotoxicity is generally considered to be associated with neuronal death, overexcitation can also have a dramatic long-term impact on neuronal structure and networks, leading to glutamatergic hypofunction [149,189]. Accordingly, it is worth noticing that ASD, substance use disorders, and psychiatric disorders share neuroplasticity alterations in glutamatergic brain areas, thus impacting mood, motivation, and cognition. Since homeostatic levels of glutamate are required for the physiological functioning of the brain [16,190,191,192,193], it should not be surprising that pharmacological strategies potentiating glutamatergic transmission were shown to restore neuroplasticity and to exert therapeutic effects in both neurodevelopmental and adult mental diseases (Table 1).

Intriguingly, among different drugs, ketamine is of particular interest because, at high doses, it can precipitate neuropsychiatric disorders (specifically, psychotic symptoms) [20,114,115], and at low doses, it has been shown to open a window of neuroplasticity [194] and is associated with rapid antidepressant, anti-PTSD, and anti-ASD actions [177,195]. Similarly, classical hallucinogens are revolutionizing the neuropsychopharmacology field, opening new paths for the management of mental disorders [182,183]. Hallucinogens are showing promising therapeutic effects not only in the management of MDD and PTSD but also for substance use diseases [113]. Again, at first glance, this might seem contradictory but not when one considers that therapeutic effects pass through the rapid regulation of neuroplasticity and connectivity [186,187].

The possibility to regulate glutamatergic transmission through mGluR modulation is attracting attention as well [196,197]. Unfortunately, to date, this approach is not yet producing particularly encouraging results, and some molecules targeting mGluRs have recently failed in clinical phases (as in the case of mGluR5 antagonists for ASD). More studies are warranted to unveil the therapeutic potential of mGluR modulators which, compared to ionotropic glutamate receptor drugs, have the advantage to offer the possibility to fine tune glutamatergic transmission. Indeed, it must be recognized that glutamatergic transmission is distributed throughout the central nervous system and plays crucial roles in several brain functions, ranging from mood to memory and cognition [3,4,5]. Thus, if on one hand glutamatergic drugs have a high therapeutic potential in diseases associated with alterations of neuroplasticity, on the other hand, they might induce dangerous effects. First of all, we cannot fail to mention that glutamatergic drugs may induce dissociative effects and may have abuse potential, as in the case of ketamine. At the same time, an excessive direct glutamatergic activation may lead to psychotic symptoms, epileptogenic effects, and excitotoxicity. Milder modulation of glutamatergic transmission, as permitted by mGluR drugs (especially in the case of positive and negative modulators) or indirect glutamatergic agents, seems to be safer but also with lower therapeutic efficacy. More studies are required to fully exploit the therapeutic potential of glutamatergic drugs in mental disorders.

**Table 1 ijms-25-06521-t001:** Therapeutic strategies targeting glutamatergic transmission for the treatment of neurodevelopmental and adult mental disorders.

Drug	Target	Disorder	Refs.
Acamprosate	NMDA receptor antagonist	SUD	[88]
Amantadine	Non-competitive NMDA receptor antagonist	SUD	[99]
Cannabidiol	TRPV1 agonist; 5-HT1A agonist; indirect CB1 and CB2 agonist	SUD	[198] Reviewed in [109,110]
d-cycloserine	Partial NMDA receptor agonist	SUD	[99]
		Schizophrenia	[132]
Dextromethadone	NMDA receptor antagonist	MDD	[179]
Dextromethorphan	Non-competitive NMDA receptor antagonist; SERT/NET blocker; Sigma σ1 receptor agonist	MDD	[181]
Hydroxynorketamine	AMPA receptor activator	MDD	[177]
Iclepertin (BI 425809)	GlyT1 inhibitor	Schizophrenia	[140]
Ketamine/S-ketamine	Non-competitive NMDA receptor antagonist	ASD	[41,42]
		SUD	Reviewed in [177]
		MDD/TRD	[176] Reviewed in [150,173,174,175,177,181]
		PTSD	[178,199]
		Other psychiatric conditions	Reviewed in [177]
R-ketamine	Non-competitive NMDA receptor antagonist	MDD	[177]
Lamotrigine	Voltage-gated sodium and calcium channel blocker	SUD	[99]
LSD	5-HT2A receptor agonist; D2/D3 receptor agonist	MDD	Reviewed in [183]
		SUD	[200]
Luvadaxistat	d-amino acid oxidase inhibitor	Schizophrenia	[133]
MDMA	Releaser and/or reuptake inhibitor of presynaptic serotonin, dopamine, and norepinephrine	PTSD	[201] Reviewed in [183]
Memantine	Non-selective NMDA receptor antagonist	ASD	[39,40]
		SUD	[100,102]
MK-801 (Dizocilpine)	NMDA receptor antagonist	SUD	[101,104]
N-acetyl-l-cysteine	mGluR2/3 agonist	ASD	[45,46]
Nitrous oxide	NMDA receptor antagonist	MDD	[181]
Pomaglumetad	mGluR2/3 agonist	Schizophrenia	[132,147]
Psilocybin	5-HT1A, 5-HT2A and 5-HT2C activator	MDD	[182] Reviewed in [183]
		SUD	[184]
Riluzole	Voltage-dependent sodium channel inhibitor; NMDA/kainate receptor antagonist	ASD	[39,40]
Sarcosine	GlyT1 inhibitor; NMDA receptor co-agonist	Schizophrenia	[137,138,139]
Sodium benzoate	d-amino acid oxidase inhibitor	Schizophrenia	[134]
Topiramate	Voltage-gated sodium channel blocker; AMPA/kainate receptor antagonist	SUD	[61,106]
Traditional antidepressants	SSRI; SNRI; TCA	MDD	[166,167]
CDPPB, ADX47273, DFB, CHPG, LSN2463359, LSN2814617	mGluR5 positive allosteric modulator	Schizophrenia	Reviewed in [145]
		SUD	[97]
TS-161, RO4995819	mGluR2/3 antagonists/allosteric modulators	MDD	Reviewed in: [177,181]
LY379268	mGluR2/3 agonist	SUD	Reviewed in [97,99]

NMDA: N-methyl-d-aspartate, TRPV1: transient receptor potential cation channel subfamily V member 1, CB1: cannabinoid receptor 1, CB2: cannabinoid receptor 2, 5-HT1A: 5-hydroxytryptamine receptor subtype 1A, ASD: autism spectrum disorder, MDD: major depressive disorder, SUD: substance use disorder, SERT: serotonin transporter, NET: norepinephrine transporter, AMPA: α-amino-3-hydroxy-5-methyl-a-isoxazolepropionate, PTSD: post-traumatic stress disorder, GlyT1: Glycine transporter type-1, TRD: treatment-resistant depression, LSD: lysergic acid diethylamide, D2: dopamine receptor 2, D3: dopamine receptor 3, MDMA: 3,4-Methylenedioxymethamphetamine, SSRI: selective serotonin reuptake inhibitor, SNRI: serotonin and norepinephrine reuptake inhibitor, TCA: tricyclic antidepressant.

## 7. Conclusions

Overall, etiopathogenetic and pharmacological studies converge in highlighting a crucial role of glutamate homeostasis and neuroplasticity in both neurodevelopmental and adult mental disorders (Figure 1). Glutamate-dependent excitotoxic processes can lead to glutamatergic dysfunctions, in turn impacting brain activity and plasticity. Neuroplastic glutamatergic drugs may have the potential to rescue pathological conditions and restore physiological neuroplasticity.

Although the clinical evidence is still limited and not always encouraging, glutamatergic modulation is representing a very interesting novel target for drug development, which is sparking research interest in different fields. Despite failures, which are an inherent part of the process, this is paving the way to novel neuroplasticity-related therapeutic strategies that could lay the foundation for next-generation pharmacotherapies to treat different brain disorders.

## Figures and Tables

**Figure 1 ijms-25-06521-f001:**
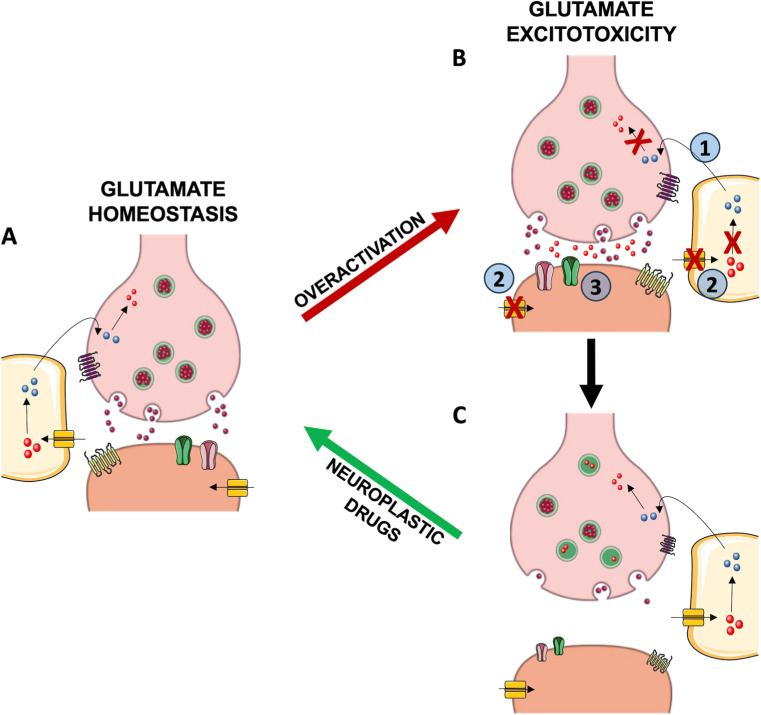
Neuroplastic drugs rescue glutamate homeostasis reverting glutamate-induced excitotoxicity and its consequences. (**A**) Physiological levels of glutamate at synapses are required for physiological brain activity, especially in those brain regions, such as the hippocampus and prefrontal cortex, implied in mood and cognitive functions and with a prevalence of glutamatergic neurons. Besides neurons, glia are also involved in the maintenance of glutamate homeostasis participating with neuronal synaptic terminals in the “tripartite synapse” to maintain balanced glutamate levels. Glutamate (red spheres) is synthesized from glutamine (blue spheres) supplied by glial cells and stored in vesicles until release throughout fusion with the presynaptic membrane. Once in the extracellular space, glutamate can bind to ionotropic (AMPA, NMDA; in pink and green) and metabotropic (mGluR; in yellow and purple) glutamate receptors at both presynaptic and postsynaptic terminals initiating several responses, including membrane depolarization, activation of intracellular messenger cascades, and modulation of local protein synthesis. Glutamate synaptic clearance is mediated by excitatory amino acid transporters (EAATs; in dark yellow) located both on neurons and astrocytes, where the reuptaken glutamate is converted into glutamine by glutamine synthetase, which is then released and ready to enter the cycle again. (**B**) The dysfunction of the mechanisms adopted to regulate synaptic glutamate levels may lead to glutamate excitotoxicity where the excessive accumulation of glutamate causes neurotoxicity and eventually cell atrophy/death. This condition may be caused by (1) impaired glutamate/glutamine recycle system, (2) dysfunctional reuptake, and (3) altered expression and activity of glutamate receptors and has been proposed as a pathogenic mechanism for neurodevelopmental and adult mental disorders. (**C**) In the long-term, the overactivation of the glutamate system may also induce impairments of neuronal structures and networks, leading to glutamatergic hypofunction. Accordingly, neurodevelopmental and adult mental disorders share neuroplasticity alterations in glutamatergic brain areas. Importantly, drugs restoring glutamatergic transmission have been demonstrated to exert therapeutic effects in these conditions together with restoring neuroplasticity. Created with smart.servier.com.

## Data Availability

No new data were created or analyzed in this study. Data sharing is not applicable to this article.

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
