# Peer review of "Glutamate-Mediated Excitotoxicity in the Pathogenesis and Treatment of Neurodevelopmental and Adult Mental Disorders"

_ijms, 2024, doi:10.3390/ijms25126521_

Round 1

Reviewer 1 Report

Comments and Suggestions for Authors

This narrative review provides a comprehensive overview of the involvement of glutamate-mediated excitotoxicity in the pathogenesis and treatment of various neurodevelopmental and mental disorders. The authors have done an excellent job in synthesizing current evidence from clinical and preclinical studies to highlight the critical role of dysregulated glutamatergic transmission in conditions such as autism spectrum disorders (ASD), substance use disorders, schizophrenia, major depressive disorder (MDD), and post-traumatic stress disorder (PTSD). The review is well-structured and organized, covering a wide range of disorders while maintaining a clear focus on the central theme of glutamate excitotoxicity. The authors provide a detailed background on the mechanisms underlying glutamate-mediated excitotoxicity, which helps in understanding the subsequent sections on specific disorders. 

While the review covers a broad range of disorders, the depth of coverage for some conditions (e.g., schizophrenia) is relatively limited compared to others. The section on therapeutic approaches could be further expanded, as it primarily focuses on rapid-acting antidepressants and psychedelics, while other potential glutamatergic targets are briefly mentioned. The review could benefit from a more critical discussion of the limitations and challenges associated with targeting glutamatergic transmission, such as potential side effects and the complexity of modulating a ubiquitous neurotransmitter system.

Author Response

This narrative review provides a comprehensive overview of the involvement of glutamate-mediated excitotoxicity in the pathogenesis and treatment of various neurodevelopmental and mental disorders. The authors have done an excellent job in synthesizing current evidence from clinical and preclinical studies to highlight the critical role of dysregulated glutamatergic transmission in conditions such as autism spectrum disorders (ASD), substance use disorders, schizophrenia, major depressive disorder (MDD), and post-traumatic stress disorder (PTSD). The review is well-structured and organized, covering a wide range of disorders while maintaining a clear focus on the central theme of glutamate excitotoxicity. The authors provide a detailed background on the mechanisms underlying glutamate-mediated excitotoxicity, which helps in understanding the subsequent sections on specific disorders. 

We thank the reviewer for his/her appreciation of our work.

While the review covers a broad range of disorders, the depth of coverage for some conditions (e.g., schizophrenia) is relatively limited compared to others. The section on therapeutic approaches could be further expanded, as it primarily focuses on rapid-acting antidepressants and psychedelics, while other potential glutamatergic targets are briefly mentioned. 

Following the suggestions of the reviewer, we expanded the sections, especially as regards glutamatergic targets and drugs.

The review could benefit from a more critical discussion of the limitations and challenges associated with targeting glutamatergic transmission, such as potential side effects and the complexity of modulating a ubiquitous neurotransmitter system.

As also suggested by the second reviewer, we expanded the discussion and future perspectives have been integrated with specific challenges associated with glutamatergic drugs.

Reviewer 2 Report

Comments and Suggestions for Authors

Dear Authors: 

The manuscript titled "Glutamate-mediated excitotoxicity in the pathogenesis and treatment of neurodevelopmental and mental disorders" by Noemi Nicosia et al, presents a very interesting review about the Glutamate-mediated excitotoxicity in neurodevelopmental and mental disorders, but certain points require further clarification:

Abstract: 

1.             The term "substance use" in the abstract is ambiguous. Does it refer to abuse? Please revise the abstract to specify that "substance use" refers to "substance abuse" to ensure that the intended meaning is clear to the readers.

Main text: 

2.          Explanation of G Protein-Coupled Metabotropic Glutamate Receptors (mGluRs). The explanation of “G protein-coupled metabotropic glutamate receptors (mGluRs), which modulate synaptic function more slowly than ionotropic receptors. Group I includes mGlu1 and 5, Group II includes mGlu2 and 3, and Group III includes mGlu4, 6, 7, and 8. Group I is coupled to Gq/G11 and activates phospholipase Cβ” needs a figure to explain it. I suggest authors include a figure that visually represents the classification and functional mechanisms of G protein-coupled metabotropic glutamate receptors. 

3.          Classification According to DSM V for Autism Spectrum Disorders. The manuscript should classify autism spectrum disorders according to DSM V (pag 100). This should include the diagnostic criteria and subcategories as defined by the DSM V, providing a clearer framework for readers.

4.          The statement “were also tested in adolescent and young adults with ASD” (pag 123) needs clarification regarding its impact. Provide specific outcomes, measures of effectiveness, and the significance of these findings to clarify their implications.

5.          The statement “rats with a history of cocaine self-administration” (lines 158-59) needs verification – is it possible? Confirm the feasibility of rats and describe the procedure. This will ensure that readers understand the experimental model used.

Thank you for considering these suggestions.

Kind regards,

Author Response

Dear Authors: 

The manuscript titled "Glutamate-mediated excitotoxicity in the pathogenesis and treatment of neurodevelopmental and mental disorders" by Noemi Nicosia et al, presents a very interesting review about the Glutamate-mediated excitotoxicity in neurodevelopmental and mental disorders, but certain points require further clarification:

We thank the reviewer for his/her appreciation of our work.

Abstract: 

  1. The term "substance use" in the abstract is ambiguous. Does it refer to abuse? Please revise the abstract to specify that "substance use" refers to "substance abuse" to ensure that the intended meaning is clear to the readers.

The term has been changed as suggested.

Main text: 

  1. Explanation of G Protein-Coupled Metabotropic Glutamate Receptors (mGluRs). The explanation of “G protein-coupled metabotropic glutamate receptors (mGluRs), which modulate synaptic function more slowly than ionotropic receptors. Group I includes mGlu1 and 5, Group II includes mGlu2 and 3, and Group III includes mGlu4, 6, 7, and 8. Group I is coupled to Gq/G11 and activates phospholipase Cβ” needs a figure to explain it. I suggest authors include a figure that visually represents the classification and functional mechanisms of G protein-coupled metabotropic glutamate receptors. 

We agree with the reviewer that the pharmacology of metabotropic (and ionotropic) glutamate receptors is very complex. Since the detailed description of their functioning is out from the scope of the present study, we prefer to cite previous comprehensive and authoritative literature on the topic. We have included a comment in the text (lines 63-64).

  1. Classification According to DSM V for Autism Spectrum Disorders. The manuscript should classify autism spectrum disorders according to DSM V (pag 100). This should include the diagnostic criteria and subcategories as defined by the DSM V, providing a clearer framework for readers.

We thank the reviewer for this remark. As suggested, we have clarified diagnostic criteria of ASD based on DSM V (lines 106-109). 106-109

  1. The statement “were also tested in adolescent and young adults with ASD” (pag 123) needs clarification regarding its impact. Provide specific outcomes, measures of effectiveness, and the significance of these findings to clarify their implications.

As requested by the reviewer, we have highlighted that clinical evidence in this context is very limited and more studies are warranted (lines 139-142).

  1. The statement “rats with a history of cocaine self-administration” (lines 158-59) needs verification – is it possible? Confirm the feasibility of rats and describe the procedure. This will ensure that readers understand the experimental model used.

We thank the reviewer for this very useful comment. Indeed, the sentence was imprecise and confounding. We have now rephrased it (lines 185-188).

Reviewer 3 Report

Comments and Suggestions for Authors

I read with interest the paper titled "Glutamate-mediated excitotoxicity in the pathogenesis and treatment of neurodevelopmental and mental disorders"

1. The review provides a comprehensive overview of glutamate-mediated excitotoxicity in various neurodevelopmental and mental disorders. However, the scope could be narrowed down to focus more deeply on specific disorders, allowing for a more detailed analysis and discussion of the most relevant studies and findings.

2. The methodology of the review should be included in the manuscript. Is this a narrative review as stated in the aim, but further details should be provided. 

3. A section on future perspectives could be expanded and divided from the conclusion. It should include more specific recommendations for future research, as identifying particular gaps in the current understanding of glutamate excitotoxicity and proposing novel experimental approaches or technologies that could address these gaps.

4. Conclusion section should be separated. 

5. Whats the impact of your research in clinical practice. This should be highlighted in the abstract. 

Author Response

I read with interest the paper titled "Glutamate-mediated excitotoxicity in the pathogenesis and treatment of neurodevelopmental and mental disorders"

We thank the reviewer for his/her appreciation of our work.

  1. The review provides a comprehensive overview of glutamate-mediated excitotoxicity in various neurodevelopmental and mental disorders. However, the scope could be narrowed down to focus more deeply on specific disorders, allowing for a more detailed analysis and discussion of the most relevant studies and findings.

As also suggested by the first reviewer, we expanded the sections, especially as regards glutamatergic targets and drugs.

  1. The methodology of the review should be included in the manuscript. Is this a narrative review as stated in the aim, but further details should be provided. 

We understand the request of Reviewer 1 regarding more methodological details, however as also recognized by the reviewer, our scope was to perform a narrative and not systematic review. Therefore, we did not use standardized inclusion and exclusion criteria, search terms, and other similar tools. Nonetheless, the literature review was rigorously performed searching for specific key words in the database PubMed. Following the Reviewer’s suggestion we have mentioned in the introduction that all the data used for our narrative review come from the PubMed database. 

  1. A section on future perspectives could be expanded and divided from the conclusion. It should include more specific recommendations for future research, as identifying particular gaps in the current understanding of glutamate excitotoxicity and proposing novel experimental approaches or technologies that could address these gaps.

As also suggested by the first reviewer, we expanded the “Future perspectives” section and added a more detailed discussion of limits and specific challenges. 

  1. Conclusion section should be separated. 

Done

  1. What’s the impact of your research in clinical practice. This should be highlighted in the abstract. 

Done